# EMG Amplitude–Force Relationship of Lumbar Back Muscles during Isometric Submaximal Tasks in Healthy Inactive, Endurance and Strength-Trained Subjects

**DOI:** 10.3390/jfmk8010029

**Published:** 2023-02-23

**Authors:** Tim Schönau, Christoph Anders

**Affiliations:** Division of Motor Research, Pathophysiology and Biomechanics, Experimental Trauma Surgery, Department for Hand, Reconstructive, and Trauma Surgery, Jena University Hospital, Friedrich-Schiller University, 07743 Jena, Germany

**Keywords:** trunk muscles, training modality, amplitude–force relationship, human

## Abstract

Previous data suggest a correlation between the cross-sectional area of Type II muscle fibers and the degree of non-linearity of the EMG amplitude–force relationship (AFR). In this study we investigated whether the AFR of back muscles could be altered systematically by using different training modalities. We investigated 38 healthy male subjects (aged 19–31 years) who regularly performed either strength or endurance training (ST and ET, n = 13 each) or were physically inactive (controls (C), n = 12). Graded submaximal forces on the back were applied by defined forward tilts in a full-body training device. Surface EMG was measured utilizing a monopolar 4 × 4 quadratic electrode scheme in the lower back area. The polynomial AFR slopes were determined. Between-group tests revealed significant differences for ET vs. ST and C vs. ST comparisons at the medial and caudal electrode positions, but not for ET vs. C. Further, systematic main effects of the “electrode position” could be proven for ET and C groups with decreasing x^2^ coefficients from cranial to caudal and lateral to medial. For ST, there was no systematic main effect of the “electrode position”. The results point towards training-related changes to the fiber-type composition of muscles in the strength-trained participants, particularly for their paravertebral region.

## 1. Introduction

The amplitude–force relationship (AFR) during isometric contractions represents the connection between surface electromyography (SEMG) amplitude and functional muscle state, i.e., the muscle force expended. SEMG is the most common approach to evaluate muscle activity/excitation in vivo. Many factors, such as fiber-type composition, contraction velocity and muscle length, are determinants of the mechanical muscle output and thus the measured SEMG signal. Therefore the functional aspects of muscles, such as fatigue [1,2], muscle co-ordination [3,4,5], or muscle recruitment [6,7,8], can be quantitatively reflected by SEMG.

The systematic correlative relationship between the measured SEMG amplitude and the respective muscle force was described by Lippold in 1952 [9], followed by Bernshtein [10], Bouisset [11] and others. Back then, a linear AFR was assumed. To our knowledge, the first systematic analysis of the AFR concerning different muscles and differently trained subjects dates back to the early 1980s, when Lawrence and De Luca identified different AFR slopes [12]. They described specific characteristics that could be mathematically determined with either linear or polynomial regressions. These regressions were muscle-specific but did not show systematic alterations by activity level or type of sport. Subsequently, Solomonow and colleagues [13] performed stimulation experiments where they found a linear AFR in a fully recruited muscle; any further force increase was only due to an increase in the firing rate. A non-linear AFR was seen if an ongoing recruitment of motor units occurred together with a firing rate increase that accounted for the force increase. They, therefore, issued “a clear warning against a direct use of SEMG to predict muscle forces” [13].

If viewed from a morphological basis, the previous studies could already demonstrate different fiber-type compositions between muscles [14,15]. Functional priorities of certain muscles have further impact on the functional cross-sectional areas (CSA) of the two main fiber types which correlated nicely with endurance and a large CSA of Type I fibers and high power output and Type II fiber’s CSA [14]. In addition, the adaption of functional fiber-type areas by specifically targeted training could be proven [16]. Our own previous investigations could confirm the known differences in the fiber-type composition of trunk muscles [17,18,19] by indirect measurements of the AFR. These were able to verify a muscle specificity with an either linear or non-linear AFR in trunk muscles with a non-linear AFR in the abdominal muscles and an almost-perfect linear AFR in the back muscles [20,21], extended by the fact that within this framework gender-specific AFR patterns could be identified [22]. Combining the facts of different fiber-type distributions in abdominal and back muscles, with a much larger CSA of Type I fibers in the back muscles [18] and a less pronounced non-linearity in women [22], the characteristics of AFR curves may be influenced by fiber-type composition. While the composition of fiber types is genetically predetermined and influenced by environmental factors [23], different training modalities alter muscle fiber types differently concerning their functional CSA.

Therefore, the current study aimed to identify the possible systematic alterations of the back muscle’s AFR slopes by using different training modalities, which may alter the functional CSA of the two main fiber types. This approach has the potential to enhance the understanding of the back muscle’s morpho-functional composition with respect to the different fiber-type areas. It therefore enables non-invasive diagnostics but additionally enables the tracing of training-, age-, or otherwise related muscular changes by electromyography. We recruited healthy volunteers who practiced either endurance or strength training at a competition-level and compared back muscle SEMG AFR curves between both groups and with a group of physically inactive participants. We expected a linear AFR in the endurance-trained and the inactive participants, whereas the strength-trained subjects were expected to show a non-linear AFR during graded defined submaximal isometric back muscle contraction tasks. As we applied a large quadratic electrode grid, independent of the training group, regional differences were expected because the investigated area included both paravertebral and more distant muscles, such as the latissimus dorsi and iliocostalis muscles. The latter ones are classified as more power-related, mobilizing muscles [24], whereas the paravertebral muscles are classified as stabilizing muscles [24,25]. Based on this and independent of the groups, we expected a lateral to medial change in AFR slopes from non-linear to a more linear characteristic.

## 2. Methods

### 2.1. Participants

For this study, 38 healthy male subjects aged between 19 and 31 years were recruited. The study cohort consisted of three groups based on their physical activity level (active and inactive) and type of training (see below). After being given information about the procedure and aim of the study, the subjects participated voluntarily and signed informed consent. The study was positively evaluated by the ethics committee of the Friedrich Schiller University Jena (2020-1844-BO).

The inclusion criteria for the two physically active groups were either endurance training (cycling or triathlon; endurance training group, ET, n = 13) or strength training (powerlifting or bodybuilding; strength training group, ST, n = 13) at a competition level (at least four training sessions per week, with a training duration of four to fifteen years). Participants who did both strength training and endurance training were not included. The subjects of the control group (C, n = 12) were not physically active at all (walking or participating in comparable activities once a week at most). The subjects’ health status was briefly checked through medical history and physical examination. Besides general health problems interfering with study participation, specific exclusion criteria were acute or recurrent back pain and deformities or surgeries of the spine. A body height outside the range of 150 to 195 cm also led to exclusion, as the subjects would not have fit into the test and training device.

### 2.2. Device and Investigation

The tasks were performed in a computerized full body tilt device for the testing and training of trunk muscles (CTT Centaur, BfMC, Leipzig, Germany), which is used in numerous studies on trunk muscles [26,27,28]. In the device, the subjects were standing upright with their hips and legs fixed, while the upper body remained unsupported. To minimize the effects of varying upper extremity positions on the investigated trunk muscles, the participants held their arms crossed in front of their chests. In the device, the subjects could be tilted from a neutral upright to a horizontal position. During the tasks, the participants simply had to stabilize their upper body along their body axis while the device was tilted at defined angles between 0°and 90°. Control of proper upright upper body position was provided through a small biofeedback monitor positioned in front of the subjects. It contained a crosshair and a moving target point that deviated from the neutral position if any net force acted on the harness, positioned over the subject’s shoulders. For this, the harness was equipped with strain gauges measuring forces in sagittal and frontal directions.

Defined forward tilt angles (0°, 5°, 10°, 20°, 30°, 45°, 60°, and 90°) were applied. Tilt angles could be converted into relative torque levels by applying the sine function and therefore corresponded to portions of 0%, 9%, 17%, 34%, 50%, 71%, 87%, and 100% of every subject’s upper body weight (Figure 1). Tilt angles were randomly applied for ten seconds each to avoid order-dependent effects.

### 2.3. Surface EMG Measurement and Analysis

Overall, 32 SEMG electrodes were applied in the lower back region on both sides utilizing a monopolar montage. For each side, a 4 × 4 quadratic electrode application scheme was used, to determine the position of the electrode (electrode position). This scheme was adjusted to each individual’s anthropometry by defining the individual L1-L4 distance as the edge length (Figure 2). The determined L1-L4 distances varied between seven and nine centimeters. Each electrode arrangement was positioned 1.5 cm laterally from the midline, using prepared electrode grids. We used a quadratic electrode grid to avoid geometric distortions due to differences in individual anthropometry. Consequently, adhering to the quadratic geometry adjustment to the lumbar vertebral distances also resulted in a varying lateral dilation of the grid, including additional muscles of the back. With the lateral border of the grid ranging from 8.5 to 10.5 cm from the midline, specifically the lateral branches of the erector spinae (i.e., iliocostalis) and latissimus dorsi muscle activations were measured by the more lateral electrodes.

For this investigation, reusable Ag-AgCl-electrodes with a diameter of six millimeters (DAGS102606, gvb geliMED, Bad Segeberg, Germany) and a hole in the middle of each electrode for electrode gel application (Electrode Cream, Care Fusion, Finland) were used. These electrodes were connected to a monopolar amplifier (ToEM16G: 10–1861 Hz (−3 dB), gain: 100, input impedance: 22 MOhm, SNR: 1.13 µV_eff_, CMRR: 91.6 dB; DeMeTec, Germany). Additionally, two conjoined electrodes (disposable Ag-AgCl electrodes, H93SG, Covidien, Neustadt an der Donau, Germany) were attached over the subjects’ anterior superior iliac spines, serving as reference electrodes whereas the ground electrode was attached at the Th11 level just over the palpable spinal process (compare with Figure 2). Electrocardiographic activation was recorded by the application of an additional bipolar input channel along the heart axis. All electrodes were applied after shaving and rubbing the examined regions with abrasive paste (SkinPure, Nihon Kohden, Tokyo, Japan). Electrodes were fixed with adhesive non-woven fabric. Special attention was paid to ensure caudal orientation of the electrode cables to avoid dislocation or levering off of the electrodes during task performance (compare with Figure 2).

Analog to digital conversion of the SEMG signals was performed with a sampling rate of 2048 per second (Tower of Measurement, DeMeTec, Langgöns, Germany, amplitude resolution: 24 bit at ± 5 V (6 nV/bit), anti-aliasing filter at 1024 Hz). SEMG data were stored on a computer (ATISArec, GJB, Ilmenau, Germany) for further analysis.

Data processing included the application of a high-pass filter at 20 Hz, a low-pass filter at 250 Hz and a notch filter at 50 Hz. ECG interferences were eliminated by using a template-based algorithm [29]. SEMG amplitude values were quantified as root mean square values, separately for each electrode and task.

As the AFR with respect to each group was the main outcome parameter, all AFR slopes were fitted applying a second-order polynomial function individually. The respective x^2^ coefficients were then calculated for every electrode position. To improve numeral display and therefore the understanding of the differences between x^2^ coefficients, all values were multiplied by 100. This accounts for all presented values.

### 2.4. Statistical Analysis

Statistical analysis of the calculated x^2^ coefficients was performed by using IBM^®^ SPSS^®^ Statistics 28 (SPSS Inc., IBM, Corp., Armonk, NY, USA). A linear mixed-effects model (LMM) was fitted to compare the effects of “group”, “side”, and “electrode position” together with interactions between these factors (required significance level *p* < 0.05). In this analysis, “group”, “side”, and “electrode position” were modelled as fixed effects with a random intercept per subject. Initially, all interactions were calculated, but for the final analysis, only the significant interactions remained in the calculation.

## 3. Results

The initial LMM calculation of the x^2^ coefficients “side” and “electrode position” together with the interaction of “group” * “electrode position” showed significant effects (all: *p* < 0.001).

As no other interactions showed significant results in addition to “side” and “electrode position”-related effects, group-specific differences that vary by electrode position were thus determined which were independent of their side. The explorative pairwise evaluation of group differences for the 16 side-independent electrode positions (now named EP as only their spatial arrangement was considered) showed that the x^2^ coefficients were systematically different between groups at the medial and caudal positions for ET vs. ST and C vs. ST, but not for ET vs. C (Table 1). As can be taken from Table 1 irrespective of proven significant differences x^2^ coefficients were always larger in the ST group.

For the main effect “side”, x^2^ coefficients were slightly larger for the right side (*p* < 0.01; left side 5.105, right side: 5.636). This was independent of the group. In both, the ET and C group x^2^ coefficients always showed decreasing values from cranial to caudal and from lateral to medial positions (Figure 3 and Figure 4). Systematic regional differences of the x^2^ coefficients were detected for the ET and C groups (always *p* < 0.001, Figure 5) but not for ST (*p* = 0.322).

## 4. Discussion

In the present study, the amplitude–force relationship of the lower back muscles was investigated during submaximal load situations. To compare the curve characteristics of the respective AFR, second order polynomials were fitted. Their x^2^ coefficients reflect the extent of non-linearity, i.e., large x^2^ coefficients indicate strong non-linearity of the AFR, whereas low values stand for a more linear curve characteristic.

In general, a non-linear AFR for ST and linear AFR for ET and C was not statistically proven. Furthermore, the detection of a possible main effect “group” in the linear mixed model did not show a significant influence regarding the x^2^ coefficients. On the other hand, in the LMM, a significant interaction was proven for “group” * “electrode position”, meaning that the different training modalities affected the x^2^ coefficients at several electrode positions. Further, in the ET and C groups systematic differences between the x^2^ coefficients were proven concerning electrode position, where the degree of non-linearity dropped from cranial to caudal and lateral to medial. In contrast, such systematic spatial differences were not proven in the ST group.

### 4.1. Recruitment Strategies

The demonstrated position- and group-specific differences of the AFR may be explained by the different recruitment strategies of the investigated muscles [13]. Published data could show that the curve characteristics of the AFR depend on how the tested muscle achieves additional power output [13]. Thus, a non-linear AFR is evident with combined recruitment of additional motor units and an increase in firing rate [13]. Linear curve slopes were proven if mainly the firing rate of the active motor units increased [13]. However, these results were obtained under stimulation conditions. The respective recruitment strategy seems to be muscle-specific [30,31]. This muscle specificity is determined by fiber composition, function/task, and the number of motor units of the respective muscle.

### 4.2. Muscle Fiber Composition

The found position-specific differences may be explained by different muscle fiber compositions and muscle fiber diameters of the investigated muscles. Muscle fiber biopsies in healthy untrained men aged 20–30 years of the longissimus and multifidus muscles at the L3 level revealed a Type I fiber content of 57–65% for both studied muscles [18,19]. Thus, there was no difference between the medially located multifidus muscle and the more laterally located longissimus muscle. Post-mortem studies by Johnson et al. revealed an average of 58% Type I fibers for the erector spinae muscle [14]. It can be postulated that due to the high stability requirements, predominantly Type I fibers occur in the medial portions of the lumbar erector spinae muscle (longissimus, multifidus, spinal, etc.), compared to more laterally located portions (iliocostalis muscle). Unfortunately, we did not find any study that has specifically investigated the iliocostalis muscle’s fiber composition. Since due to their anatomical location latissimus dorsi and iliocostalis muscles are both used less for stability but more for movement execution [32], an elevated Type II fiber content can be assumed. In the study by Johnson et al. a ratio of 50% Type I fibers could be demonstrated in the latissimus dorsi muscle [14]. The more heterogeneous the muscle fiber composition of a muscle and the larger the Type II muscle fiber CSA, the more non-linear the AFR will be [33]. Our own previous studies with untrained subjects showed a non-linear AFR for abdominal muscles, whereas an almost ideal linear AFR was found in the back muscles [20]. According to the literature, abdominal muscles show a Type I fiber content of 46–58% [14,17], whereas back muscles contain 57–65% Type I fibers [14,24,25]. This fact and the current results support the hypothesis of a fiber-type-dependent curve characteristic of the AFR. Type II muscle fibers exhibit a higher density of Na^+^ channels, faster action potentials, and a higher resting membrane potential [33]. During isometric contractions, motor units are recruited in an ordered sequence: firstly, fibers with a low shortening velocity (Type I) and secondly, fibers with a higher shortening velocity (Type II) [34,35]. This results in disproportionately larger potentials at higher force levels, which are generated primarily by motor units of Type II fibers, producing an exponential and therefore non-linear AFR. This cannot be demonstrated in muscles with a predominant Type I portion [31]. In the current investigation, all subjects showed a non-linear AFR in the most lateral and cranial region of the investigated region, where mainly latissimus dorsi and iliocostalis muscles are to be found. The respective AFR can consequently be explained by the significantly more heterogeneous fiber composition of these muscles.

### 4.3. Training Associated Effects

The ST participants showed no group-specific systematic differences in x^2^ coefficients concerning electrode position, but consistently large x^2^ coefficients that were significantly different from the two other groups at the caudal and medial positions. Although the individual proportion of Type I and Type II fibers is genetically predetermined and influenced by environmental factors [23], fiber composition and CSA can be altered by training evoked adaptation [36]. Studies have shown that pure endurance training does not affect functional muscle fiber CSAs [37,38]. Strength training and combined endurance and strength training both increase Type II fibers CSAs by 13–18%. Isolated strength training additionally increased Type I fiber CSA [38]. Although we did not take muscle biopsies, the mentioned training-induced effects can also be assumed in our active groups because all of our trained subjects performed years of intense training in their sports. As already mentioned, systematic differences between groups could only be determined for the most caudal and medial positions. The more lateral electrode positions contain information of lateral branches of the erector spinae, i.e., iliocostalis and also latissimus dorsi muscles, which showed a clear non-linear AFR also in the ET and C groups and therefore precluded any systematic differences between the groups. It is reasonable to assume that the described training-associated effects on muscle fibers in the ST group caused areas with assumable linear AFR (medial and caudal) to change to a non-linear AFR. Medial and caudal linear AFRs were detected in the ET and C groups, but not in the ST group. Thus, the position-specific effects found in the ET and C groups were abolished by strength training, which may also result in the non-systematic x^2^ coefficients distribution found in the ST group in the per-group analysis. This explains the found systematic differences in the group-electrode-interaction analysis in the group comparisons (Table 1). The described remaining non-linearity of the AFR in the medial muscle components of the ST participants ensured that group-specific differences were most pronounced in this area.

We found significant side effects with a higher non-linearity on the right side (compare with Figure 3). This may be explained by the fact that, except for two subjects (one ET, one C), only right-handed individuals participated in our study. The contralateral side of the back muscles is known to be more fatigue-resistant due to their predominant stability function when working with the right upper extremity [39]. Therefore, a higher proportion of Type I fibers compared to the ipsilateral side seems to be likely. This may result in an overall higher non-linearity on the right side with respect to the contralateral side.

### 4.4. Clinical Implications

Our trunk and particularly our back muscles play a key role in mobilizing and stabilizing the spine. Mainly impaired back muscle force control [40] or corrupted coordination causing subfailure injuries [41] are considered as relevant causes for back pain. In contrast, differences in maximum force production seem to be of inferior importance concerning the development of lower back pain [42]. In this respect, isolated strength training does not seem to be the decisive factor for the prevention of lower back pain. Moreover, if during sustained muscle activity, not only fatigue-related muscle force, but also co-ordination is corrupted [43], then acute episodes of lower back pain are more likely to occur. Therefore, the presented data are another strong argument for the application of functionally oriented physical training in back pain patients to prevent episodes of acute back pain and also to restore the necessary equilibrium between the stability and mobility of the spine. For this, multimodal rehabilitation programs were developed which not only aim to improve the physical state of the patients, but also add educative parts in order to restore or even built more self-reliant movement patterns, based on regained physical possibilities without fear of exercise. With respect to the treatment of lower back pain, not only are functional parameters known to be important, but also psychological characteristics [44], but these were not the focus of the current investigation.

### 4.5. Limitations

As no muscle biopsies were taken, the drawn conclusions about the fiber composition remain somehow speculative. On the other hand, as could be proven elsewhere during an endurance test exploring the same population [45] the participants of the ST group were more prone to back muscle fatigue in comparison to the other two groups. This indirectly proves the correctness of the assumed differences of the functional fiber-type areas between the groups.

The applied submaximal static tests were performed in a specific test and training device, which is not available everywhere. Besides this drawback, using the device we could apply exactly defined loads of the mentioned portions from 9% to 100% upper body weight to all participants without any uncertainty regarding the correct measurement of each participant’s upper body weight [28] by simply tilting them at the respective tilt angles in the sagittal plane.

The electrodes were applied using a monopolar montage, which is much more prone to cross-talk than bipolar montages [46]. We decided to use this montage, since therewith we could align the electrodes according to each individual’s morphology. The natural drawback of this individual alignment was that inter-electrode distances vary between subjects, and therefore prohibit bipolar signal calculation.

## 5. Conclusions/Suggestions for Future Research

Isolated strength training is accompanied by relevant changes in the AFR of back muscles, especially for their paravertebral portions. These modifications can be attributed to sport-specific changes in fiber composition. As part of the evolutionary development of upright posture and the associated use of the arms for load manipulation, based on their fiber composition paravertebral muscles have become more specialized for postural tasks. Therefore, remodeling processes with an increased Type II fiber area could have rather detrimental consequences in the long term.

As the results showed clear differences between the groups, training or rehabilitation programs may be accompanied by SEMG measurements to monitor the induced changes in functional capacity. The effects of different training methods could thereby be evaluated. As only submaximal forces were applied, the results are mostly independent from the subject’s motivation.

## Figures and Tables

**Figure 1 jfmk-08-00029-f001:**
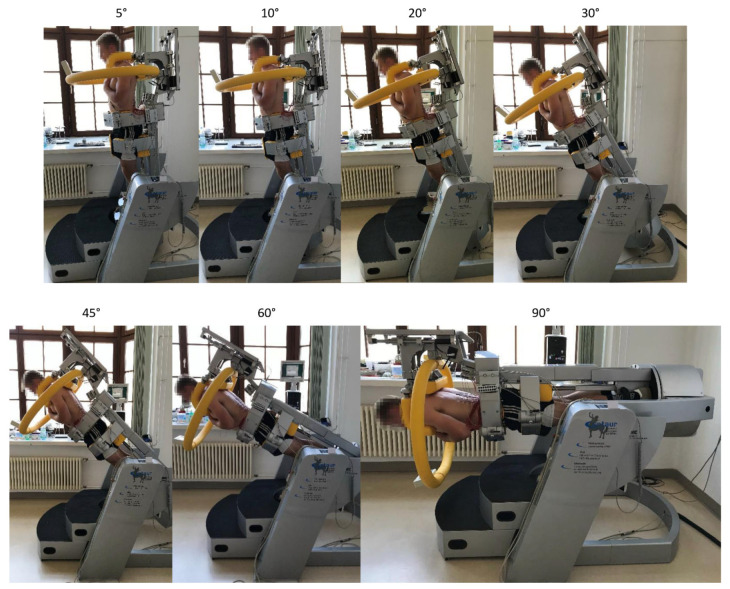
Applied tilt angles for the determination of the EMG amplitude–force relationship. Note that the upper body is always aligned along the body axis and the arms are held crossed in front of the chest.

**Figure 2 jfmk-08-00029-f002:**
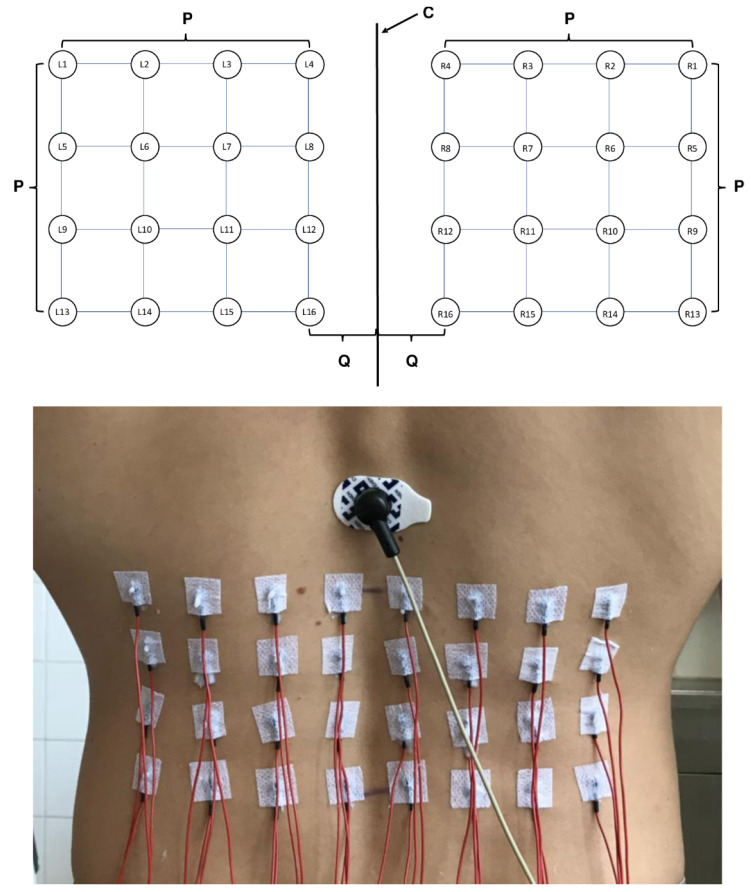
Top: Schematic display of the electrode montage. C: center line, P: edge length (adjusted to individual L1-L4 distance), Q: fixed distance of 1.5 cm from the center line. Electrode positions are indicated (L for left-sided, R for right-sided positions). Please note that electrode numbers are mirrored with respect to their lateral to medial orientation. Bottom: Electrode application as it appears on a participant.

**Figure 3 jfmk-08-00029-f003:**
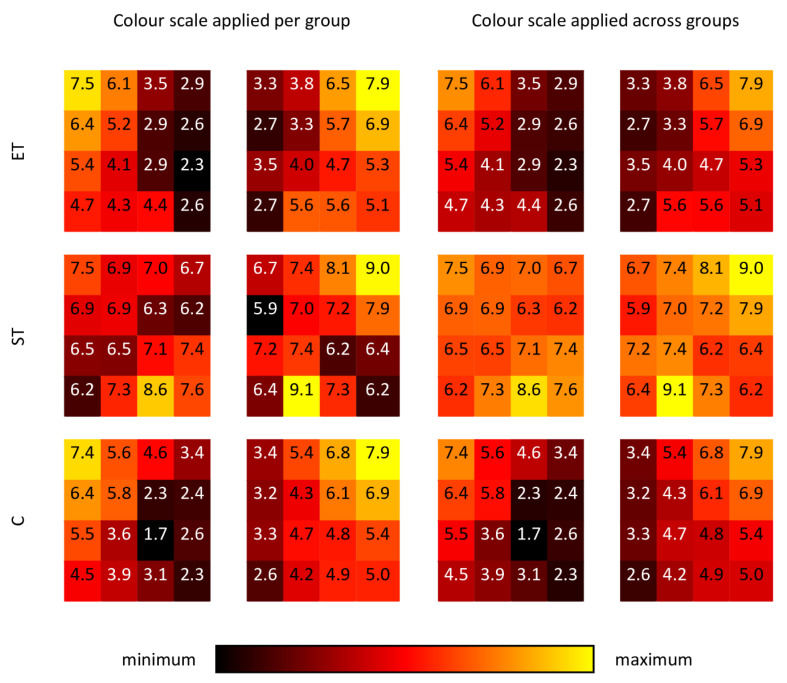
Map-like representation of mean x^2^ coefficients per electrode position, separately for each group (note that values were multiplied by 100 for improved numeral display). The minimum to maximum color scale is either separately applied for each group (**left side**) or together for all groups (**right side**).

**Figure 4 jfmk-08-00029-f004:**
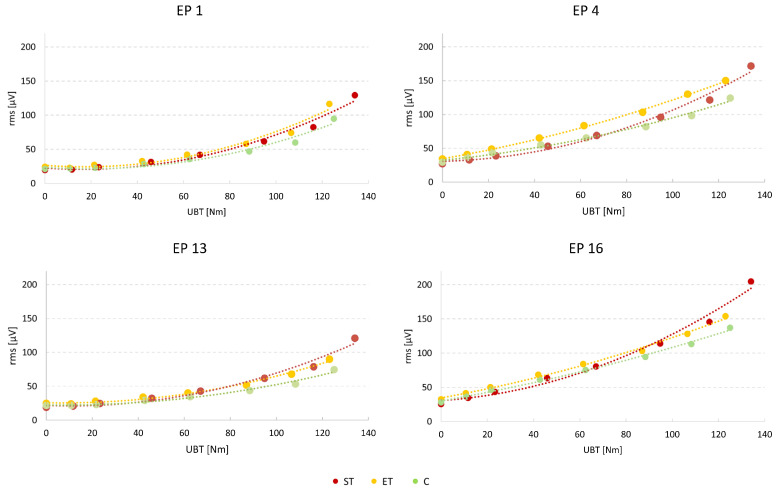
Examples of mean AFR slopes per group at selected positions at the left side. UBT: upper body torque.

**Figure 5 jfmk-08-00029-f005:**
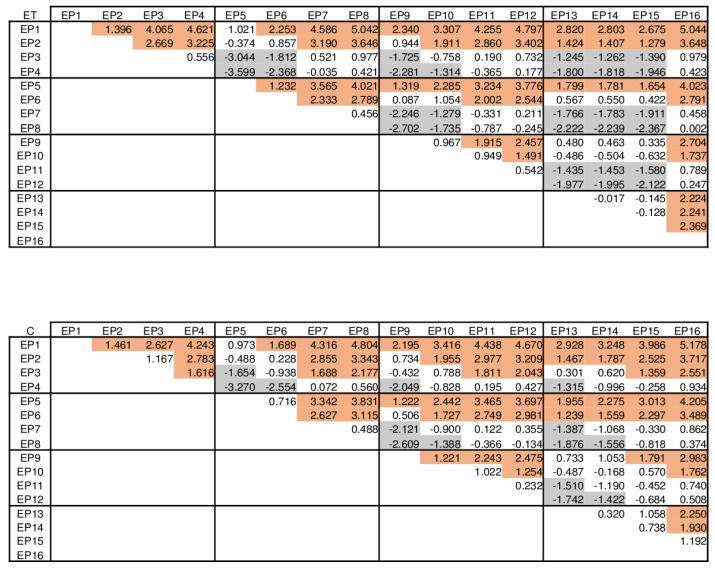
Pairwise comparisons of the x^2^ coefficients for ET (**upper panel**) and C (**lower panel**) groups between side-independent electrode positions (EP). Values correspond to the mean differences of each comparison. Significant (least significant difference) differences between x^2^ coefficient values are indicated by shadings. Brown shading: line heading position > column heading position; grey shading: line heading position < column heading position.

**Table 1 jfmk-08-00029-t001:** Pairwise comparisons of the x^2^ coefficients between groups per side-independent electrode position (EP).

	ET vs. ST			C vs. ST			ET vs. C		
		95% CI			95% CI			95% CI	
	Mean Diff.	Lower Border	Upper Border	Mean Diff.	Lower Border	Upper Border	Mean Diff.	Lower Border	Upper Border
EP1	−0.580	−4.388	3.228	−0.652	−4.629	3.326	0.072	−3.906	4.049
EP2	−1.168	−4.976	2.641	−1.304	−5.282	2.673	0.136	−3.841	4.114
EP3	−3.543	−7.351	0.265	−2.176	−6.154	1.801	−1.366	−5.344	2.611
EP4	−3.611	−7.419	0.197	−3.305	−7.283	0.672	−0.306	−4.283	3.672
EP5	−0.683	−4.491	3.126	−0.706	−4.683	3.272	0.023	−3.954	4.001
EP6	−1.598	−5.406	2.210	−1.106	−5.083	2.872	−0.493	−4.470	3.485
EP7	−3.546	−7.354	0.262	−3.347	−7.325	0.630	−0.199	−4.176	3.779
EP8	−3.393	−7.201	0.416	−3.226	−7.204	0.751	−0.167	−4.144	3.811
EP9	−1.064	−4.872	2.744	−0.990	−4.968	2.987	−0.073	−4.051	3.904
EP10	−1.940	−5.748	1.868	−2.120	−6.098	1.857	0.181	−3.797	4.158
EP11	−3.807	−7.615	0.001	−4.061	−8.039	−0.084	0.254	−3.723	4.232
EP12	−4.373	−8.182	−0.565	−4.318	−8.295	−0.340	−0056	−4.033	3.922
EP13	−1.286	−5.094	2.522	−1.465	−5.443	2.512	0.179	−3.798	4.157
EP14	−2.405	−6.213	1.403	−2.922	−6.899	1.056	0.517	−3.461	4.494
EP15	−3.823	−7.631	−0.015	−5.205	−9.183	−1.228	1.382	−2.595	5.360
EP16	−4.341	−8.150	−0.533	−4.547	−8.524	−0.569	0.206	−3.772	4.183

negative values: first group < second group. Shaded cells: *p* < 0.05 (adjustment for multiple tests: least significant difference).

## Data Availability

All data are available upon request.

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
