# Peer review of "EMG Amplitude–Force Relationship of Lumbar Back Muscles during Isometric Submaximal Tasks in Healthy Inactive, Endurance and Strength-Trained Subjects"

_jfmk, 2023, doi:10.3390/jfmk8010029_

Round 1

Reviewer 1 Report

The Article  describes a study investigating the relationship between the cross-sectional area of Type II muscle fibers and the degree of non-linearity of the EMG amplitude to force relationship (AFR) in the back muscles. The study involved 38 male subjects who performed either strength or endurance training or were physically inactive. The results showed significant differences between the endurance training and strength training groups as well as the control group and the strength training group, but not between the endurance training and control groups. The results suggest that strength training may alter the fiber type composition in the paravertebral region of the back muscles.

Overall, the Article provides a clear and concises objective, methodology, and results. The language used is appropriate for a scientific article and the results are presented in a clear and concise manner. However, it would be helpful to provide some context and explain the significance of the findings in terms of the impact on future research or practical applications.

Author Response

The Article  describes a study investigating the relationship between the cross-sectional area of Type II muscle fibers and the degree of non-linearity of the EMG amplitude to force relationship (AFR) in the back muscles. The study involved 38 male subjects who performed either strength or endurance training or were physically inactive. The results showed significant differences between the endurance training and strength training groups as well as the control group and the strength training group, but not between the endurance training and control groups. The results suggest that strength training may alter the fiber type composition in the paravertebral region of the back muscles.

Overall, the Article provides a clear and concises objective, methodology, and results. The language used is appropriate for a scientific article and the results are presented in a clear and concise manner. However, it would be helpful to provide some context and explain the significance of the findings in terms of the impact on future research or practical applications.

Answer

We have now expanded the Conclusions heading to " Conclusions/Suggestions for future Research" and also added a respective paragraph, highlighting the impact of the findings according future research (L351-355).

Reviewer 2 Report

jfmk-2207206

Reviewer comments

In the submitted manuscript, the author(s) studied the EMG amplitude to force relationship of back muscles to check the hypothesis that different training modalities can alter this relationship. Three groups were examined: endurance trained, strength/power trained, and inactive young males. Forces were measured using a tilting apparatus, while EMGs of the lower back muscles were recorded with a monopolar 4x4 quadratic electrode. Results revealed that between group differences were observed at medial and caudal electrode positions. In addition, no main effect of electrode position was evident. It is concluded that the findings suggest changes of fibre type composition in the examined muscle group.

A number of topics need to be addressed, as mentioned below.

General comments:

  • Further elaboration is required in the Introduction to provide a basis for the second hypothesis of the study: Why regional differences are expected? Why the quadratic electrode grid is the preferred method to collect the data to check the hypotheses?
  • Specify the targeted muscles in the Methods section.
  • The assumption made in L261-264 also reflects a major concern as it comprises a fundamental limitation of the study and a generalization about the fibre type composition that is a basis for one of the comparisons made. Further depiction of the within group distribution could clarify the homogeneity of the data and thus provide evidence that group membership satisfied the fibre type difference basis for forming the experimental groups.

Specific comments:

  • L11: In this
  • L16: suggested to change to 4x4.
  • L27-34: merge these two paragraphs.
  • L56-56: provide references for this statement.
  • L79: it is recommended to use other term for ‘present’.
  • L155-162: state the significance level for the statistical analysis.
  • L201-209: use ‘was’ instead of ‘could’.
  • L212-214: provide references for this statement.
  • L280: add the limitations of the study and suggestions for future research.
  • L310-379: provide the journal abbreviated title for references #2, #8, #9, #14-18, #20, #25, #28, #30-33.

Author Response

In the submitted manuscript, the author(s) studied the EMG amplitude to force relationship of back muscles to check the hypothesis that different training modalities can alter this relationship. Three groups were examined: endurance trained, strength/power trained, and inactive young males. Forces were measured using a tilting apparatus, while EMGs of the lower back muscles were recorded with a monopolar 4x4 quadratic electrode. Results revealed that between group differences were observed at medial and caudal electrode positions. In addition, no main effect of electrode position was evident. It is concluded that the findings suggest changes of fibre type composition in the examined muscle group.

A number of topics need to be addressed, as mentioned below.

General comments:

  1. Further elaboration is required in the Introduction to provide a basis for the second hypothesis of the study: Why regional differences are expected? Why the quadratic electrode grid is the preferred method to collect the data to check the hypotheses?

Answer

We have added the respective information (L49-54, L77-83)

  1. Specify the targeted muscles in the Methods section.

Answer

The respective information has been added (L133-139)

  1. The assumption made in L261-264 also reflects a major concern as it comprises a fundamental limitation of the study and a generalization about the fibre type composition that is a basis for one of the comparisons made. Further depiction of the within group distribution could clarify the homogeneity of the data and thus provide evidence that group membership satisfied the fibre type difference basis for forming the experimental groups.

Answer

We have provided the between electrode position tests in the results section (Figure 4), which clearly document the mentioned lateral to medial and cranial to caudal systematic in the ET and C groups. At this point we would therefore prefer to leave the data as they are. We have elaborated the respective matter more in detail in the discussion (L285-290).

Specific comments:

  • L11: In this

Done

  • L16: suggested to change to 4x4.

Done

  • L27-34: merge these two paragraphs.

Done

  • L56-56: provide references for this statement.

Done

  • L79: it is recommended to use other term for ‘present’.

Done

  • L155-162: state the significance level for the statistical analysis.

Done

  • L201-209: use ‘was’ instead of ‘could’.

Done

  • L212-214: provide references for this statement.

Done

  • L280: add the limitations of the study and suggestions for future research.

Limitations appear in the following section (L325-342). Suggestions for future research were added (L351-355)

  • L310-379: provide the journal abbreviated title for references #2, #8, #9, #14-18, #20, #25, #28,

Done

Round 2

Reviewer 2 Report

  • In the resubmitted manuscript, the author(s) addressed adequately all concerns mentioned in the initial round of review.
  • A depiction of the intergroup AFR traces could add to the manuscript.

Author Response

  • In the resubmitted manuscript, the author(s) addressed adequately all concerns mentioned in the initial round of review.

Thank you for your positive evaluation of the revised manuscript

  • A depiction of the intergroup AFR traces could add to the manuscript.

We have now added a new figure 4, containing exemples of AFR traces for traceability of the results
